# Endothelial Dysfunction in Diabetes Mellitus: New Insights

**DOI:** 10.3390/ijms241310705

**Published:** 2023-06-27

**Authors:** Michal Dubsky, Jiri Veleba, Dominika Sojakova, Natalia Marhefkova, Vladimira Fejfarova, Edward B. Jude

**Affiliations:** 1Diabetes Centre, Institute for Clinical and Experimental Medicine, 14021 Prague, Czech Republic; jivb@ikem.cz (J.V.); sojd@ikem.cz (D.S.); maqn@ikem.cz (N.M.); vlfe@ikem.cz (V.F.); 2First Faculty of Medicine, Charles University, 14021 Prague, Czech Republic; 3Diabetes Center, Tameside and Glossop Integrated Care NHS Foundation Trust, Ashton-under-Lyne OL6 9RW, UK; edward.jude@tgh.nhs.uk; 4Department of Endocrinology and Gastroenterology, University of Manchester, Manchester M13 9PL, UK

**Keywords:** endothelial dysfunction, diabetes mellitus, vascular health, arterial stiffness

## Abstract

Endothelial dysfunction (ED) is an important marker of future atherosclerosis and cardiovascular disease, especially in people with diabetes. This article summarizes the evidence on endothelial dysfunction in people with diabetes and adds different perspectives that can affect the presence and severity of ED and its consequences. We highlight that data on ED in type 1 diabetes are lacking and discuss the relationship between ED and arterial stiffness. Several interesting studies have been published showing that ED modulates microRNA, microvesicles, lipid levels, and the endoplasmatic reticulum. A better understanding of ED could provide important insights into the microvascular complications of diabetes, their treatment, and even their prevention.

## 1. Introduction

The endothelium plays a crucial role in maintaining vascular homeostasis by regulating blood flow, maintaining vascular tone, and preventing thrombosis [1]. The endothelial layer, of which the endothelial cell is the main component, has emerged as the key regulator of vascular equilibrium, in that it not only has a barrier function but also acts as an active signal transducer for circulation that can modify the vessel wall quality.

The main function of the endothelial cells (ECs) is protection from blood coagulation and transcytosis. Therefore, the alteration of these functions can result in impaired function in ECs, which can lead to atherosclerosis and, subsequently, thrombosis or embolisms [2]. Biomechanical forces, including shear stress from disturbed blood flow, also activate the endothelium, which leads to an increase in vasomotor dysfunction and promotes inflammation by upregulating pro-atherogenic genes [3].

Lipids (in particular, low-density lipoproteins—LDL) play a significant role in regulating these functions by reducing the bioavailability of nitric oxide (NO) and activating pro-inflammatory signaling pathways such as nuclear factor kappa B (NFK-B). The repair process of endothelial lesions depends on the expression of vascular endothelial growth factor A (VEGFA) and is also influenced by turbulent blood flow within the vasculature [4].

Increasing damage to the ECs induces additional synthesis and the secretion of components of basement membranes and the augmentation of the synthesis of the Von Willebrand factor. 

When endothelial dysfunction (ED) occurs, there is reduced NO bioavailability (a potent vasodilator), increased oxidative stress, and decreased production of endothelium-derived relaxing factors [5]. This leads to a decrease in the ability of blood vessels to relax and dilate, resulting in impaired blood flow, increased blood pressure, and a higher risk of cardiovascular disease. ED can be a consequence of several factors, including diabetes, hypertension, hypercholesterolemia, smoking, and obesity [6,7].

## 2. ED in Type 1 and Type 2 Diabetes

Endothelial function is impaired in a lot of tissues in people with diabetes, especially via reduced NO production, increased endothelin-1, and the alteration of adipokine secretion. ED can contribute to diabetic complications, and therefore, the endothelium could be a potential target for the prevention of complications in DM [8]. In type 1 diabetes mellitus (T1DM), uncontrolled hyperglycemia, glycemic variability, and low concentrations of endogenous insulin are the key factors responsible for the pathogenesis of ED [9], which is in contrast with type 2 diabetes mellitus (T2DM), where the most important factors are insulin resistance, hyperglycemia, and dyslipidemia. It has been shown that the pericapillary basement membrane is 38.5% and 45.5% thicker (*p* < 0.05) in people with type 2 diabetes mellitus (T2DM) as a result of peripheral arterial disease than in healthy subjects and people with arterial hypertension, respectively. The proportion of capillaries with disrupted basement membranes between pericytes and EC is higher (*p* < 0.05) in people with arterial hypertension (33.2%) and those with T2DM (38.7%) than in control subjects [10].

The pathogenesis of ED in T1DM and T2DM involves oxidative stress, chronic inflammation, and low NO production. Takeda G protein-coupled receptor 5 (TGR5) is a G protein-coupled receptor that is present in many organs and tissues and is widely expressed in almost all types of EC. A recent study showed that TGR5 mediates beneficial effects in both types of DM and regulates various molecules that cause DM-associated ED [11]. It has also been shown that adolescents with T1DM have impaired baseline conduit artery NO-dependent vasodilation in response to shear stress [12]. Factors that positively and negatively influence endothelial cells are shown in Figure 1.

MicroRNAs (miRs) play a very important role in the pathogenesis of cardiovascular disease. They are also associated with inflammation and immune response in humans [13]. The miR-200c isoform is present in ECs (especially in the aorta), cardiomyocytes, and vascular smooth muscle cells. This isoform was demonstrated to increase in an animal model of diabetic rats with cardiomyopathy. Another important miR-miR-34a promotes senescence in the vascular endothelium and has been shown to be a mediator of vascular pathologies in people with DM [14]. miR-126, which is specific to endothelial cells, promotes angiogenesis in response to angiogenic growth factors [15] and has been demonstrated to be reduced in patients with diabetic retinopathy [16].

In T1DM, there is chronic inflammation with increased levels of IL-7, IL-8, TNF-alpha, and vascular endothelial growth factor-C (VEGF-C) [17]. Patients with T1DM have decreased levels of endothelial progenitor cells and, therefore, have an increased risk of cardiovascular disease (CVD). miR-200c is mostly associated with the PI3K-Akt and VEGF signaling pathways.

Bakhashab et al. demonstrated that miR-200c-3p was significantly downregulated in the peripheral blood mononuclear cells of T1DM patients and negatively correlated with HbA1c, IL-7, VEGF-C, and sVCAM-1 [18]. On the other hand, this microRNA positively correlated with a number of CD34 and VEGFR-2+ cells, as well as proangiogenic cells. These results support the theory of the anti-angiogenic effect of miR-200c. The authors concluded that miR-200c-3p could be an early marker of subclinical CVD in T1DM patients because it is correlated with the markers of vascular health and diabetes control.

Circulating angiogenic factors, especially fibroblast growth factor 2 (FGF-2), play an important role in vascular inflammation and atherosclerotic lesions, mostly via the stimulation of intimal growth, angiogenesis within plaque, and the proliferation of smooth muscle cells [19].

Schönborn et al. demonstrated that patients with more severe diabetic foot ulcers had significantly higher FGF-2, and therefore, this growth factor could be a potential marker of ulcer progression [20].

## 3. Methods of Assessing ED

There are many methods and parameters aimed at evaluating arterial stiffness (AS), and the most frequently used are pulse wave velocity (PWV), change in artery diameter due to pressure changes, augmentation index (AI), pulse pressure, and imaging using MRI or CT.

AS and ED can be considered two different sides of vascular disease, in that some connections can be found between them [21]. NO plays an important role in arterial vasodilatation, and therefore, pharmacological drugs that improve ED, especially angiotensin-converting enzyme inhibitors or statins, can also decrease the level of AS. Evidence from both animal and human studies demonstrated that the endothelium is an important regulator of AS via its structure and function [22].

The first option to assess ED is flow-mediated dilatation (FMD), where ED is characterized by less vasodilatation (reduced FMD) in the brachial artery. FMD is assessed using ultrasound to measure changes in brachial artery diameter in response to increased flow after a period of vascular occlusion caused by a blood pressure cuff and is highly dependent on NO bioavailability [23]. Endo-PAT 2000 is a newer, non-invasive technology for measuring ED. In ED, the reactive hyperemia index (RHI) is low, and the pulse amplitude is high. Endo-PAT also assesses the peripheral augmentation index (PAT-AI). Bonetti et al. reported that an RHI of <1.35–1.49 is indicative of coronary ED in adults [24]. FMD is indicative of large vessel reactivity, whereas RHI is that of the small vessels, which may account for the challenges in comparing the two parameters. Wilk et al. reported that RHI correlated with FMD (r = 0.35, *p* < 0.01); however, there are other studies that have not reported an association between FMD and RHI [25].

Non-invasive pneumatic probes placed on both index fingers can continuously record pulse wave amplitude. A blood pressure cuff is placed over the brachial artery and inflated to occlude blood flow, and the response after deflation is recorded. The RHI is measured following this mini-ischemic stress to the vessel. The pulse wave amplitude (PWA) is measured and computes an RHI result automatically. The RHI is calculated as the ratio of the average PWA divided by the average amplitude during the equilibration period. To compensate for any systemic changes, this ratio is normalized to a concurrent signal from the contralateral finger.

Hayden et al., in their recent study, reported that Endo-PAT was an easy test to conduct and well-tolerated by children and adolescents with T1DM, and it could be performed at the point of care [23]. Two studies reported lower RHI results in T1DM [26,27]. Shah et al. reported that T2DM patients had greater vascular thickness and stiffness and more severe ED compared with obese and lean children [28].

In conclusion, Endo-PAT is a useful technology for measuring ED in pediatric subjects who suffer from T1DM or T2DM.

## 4. Arterial Stiffness in Patients with Diabetes

There are a lot of data about the relationship between arterial stiffness (AS) and T2DM. Arterial walls in people with T2DM are usually highly calcified, especially in those with long-duration diabetes, and these calcifications are distributed mostly in arteries below the knee and foot. Patoulias et al. researched the cardioprotective role of SGLT2 inhibitors, and they concluded that the role of these drugs in AS is limited and controversial [29]. Another important finding was that patients with recently diagnosed T2DM and no cardiovascular risk factors had higher AS compared with healthy controls and, therefore, probably had premature vascular aging [30]. Some papers have also emphasized the importance of the early diagnosis of AS in T1DM and its clinical value in the prevention of CVD [31].

An association between AS and cardiovascular risk In T1DM patients has also been reported. The increased AS could be due to insulin resistance, collagen increase as a consequence of inadequate enzymatic glycation, and endothelial and autonomic dysfunction (Figure 2). On the other hand, there is not much evidence for the use of pharmacotherapy (ACE inhibitors, angiotensin II receptor antagonists, or SGLT2 inhibitors) in the prevention of AS, but, on the other hand, newer pharmacological agents were recently assessed for the treatment of CVD [32]. Further research in this area is required to better understand the pathophysiology of AS in T1DM.

Love et al. demonstrated in their recent study on 41 young individuals with T1DM that age, sex, and FMD were unique predictors of carotid–femoral PWV (cfPWV) [33]. They also showed that men had significantly higher cfPWV compared with women, but other macrovascular parameters, such as AI and FMD, were not different between these genders.

Patients who develop T1DM in the first decade of their lives have a very high risk of cardiovascular disease compared with patients developing T1DM after the second and third decades of life [34]. Fasting hyperglycemia and glycated hemoglobin (HbA1c) are not always good predictors of carotid artery wall thickening, which increases in T1DM patients [35,36]. A recent report including a large population of patients with T2DM described the significant association between time in range (TIR) and carotid intima–media thickness (cIMT) [37]. TIR was defined as the level of glucose between 3.9 and 10 mmol/L measured using continuous glucose monitoring sensors [38]. Different authors, on the other hand, found no association between TIR and IMT in women, probably because they were healthier than men in the study. Clinical studies have demonstrated a significant 8–13% higher cardiovascular risk for each percentage point decrease in brachial artery FMD [39]. Cutruzolla et al. evaluated whether carotid thickening and ED are associated with TIR in patients with T1DM. Furthermore, the study evaluated the association between cIMT, FMD, and the time below range (TBR), the time above range (TAR), mean daily glucose, and glycemic variability. The mean cIMT of the right common carotid artery (CCA) and the mean–maximal cIMT of CCA were significantly higher and FMD significantly lower in T1DM than the control group. The regression analysis did not reveal any statistically significant association between TIR, the mean–maximal cIMT of CCA, and the FMD of the brachial artery after ischemia and handgrip exercises. No association was detected between the thickness of the CCA and the function of the brachial artery and the TBR, the TAR, mean daily glucose, and CV. HbA1c was also not significantly associated with IMT or FMD. Conversely, a significant association was found between the mean–maximal IMT, FMD, age, and diabetes duration. Hypoglycemia and glycemic variability were not associated with FMD and IMT. This study showed that TIR could not help identify T1D patients with early atherosclerosis.

## 5. Role of Microvesicles in ED

Extracellular vesicles (EVs) are a group of nano-sized vesicles released into the circulation system by endothelial cells and platelets, usually upon activation and apoptosis [5]. The usual sources of EVs are body fluids or cell lines from conditioned media (usually adipose-tissue-derived or bone-marrow-derived mesenchymal stem cells). EVs also influence ED in patients with T2DM. Fluitt et al. suggested that miRNAs contained in EVs could be used as markers of ED in patients with T2DM, especially in those with microvascular complications, such as diabetic kidney disease [14]. EVs derived from adipose tissue mesenchymal stem cells have been shown to decrease the apoptosis of endothelial and smooth muscle cells in the corpus cavernosum of diabetic rats [40].

Medium-to-large-sized EVs released through membrane budding are also called microvesicles (MVs). MVs are identified as particles of <0.9 μm in size, whereas the general size of an EV is very heterogenous, between 30 nm to 10 μm. EVs are able to carry miRNA between different cells. Moreover, miRNAs in EVs can influence cellular morphology, usually by modulating its translational profile [41].

High-mobility group box-1 protein (HMGB1) is a nuclear DNA-binding protein released either passively upon apoptosis or necrosis or actively upon the activation of monocytes. It is also stored in platelets and is released into the circulation system upon their activation [42]. Bergen showed that, in 236 patients with T1DM and 100 healthy matched controls, T1DM, regardless of the presence of microangiopathy, was associated with a higher number of platelet MVs. Moreover, patients with T1DM had significantly higher levels of HMGB1+ platelet MVs and endothelial MVs compared with healthy controls, which could potentially aggravate ED in these patients and contribute to future CVD.

## 6. The Importance of Lipids

In a case–control study, Alkaabi et al. investigated the relationship between plasma lipids and inflammatory and ED biomarkers in young patients with and without T1DM [43]. The study included 158 patients with T1DM and 157 healthy controls. Anthropometric data, BMI, lipid profiles (high-density lipoprotein—HDL, triglycerides, ApoA, and ApoB), liver enzymes (GGT and ALT), and inflammatory (IL-6, adiponectin, TNF-α, hs-CRP) and ED biomarker levels (ICAM-1, VCAM-1, selectin, and endothelin-1) were measured in all participants. There was a significant association between increases in lipid profiles, liver enzymes, inflammatory markers, and ED markers in T1DM patients compared with the controls. Among the biomarkers studied, ApoA, ApoB, and TC were significantly increased in T1DM patients compared with the controls. The study showed an extensive correlation between different biomarkers in patients with T1DM in comparison with age-and sex-matched controls. This study demonstrated that T1DM patients are exposed to various cardiovascular risk factors because of increased vascular inflammation and the presence of ED.

A recently published review described new insights into HDL structure and function in both T1DM and T2DM [44]. This paper summarized the understanding of the potential causal relationships between HDL and the pathogenesis of vascular inflammation and premature atherosclerosis on the one hand and between HDL and glucose homeostasis on the other. According to recent findings, in young T1DM patients with early renal impairment and high inflammatory scores, both HDL antioxidative activity and endothelial vasodilatory function were impaired, revealing a critical link between HDL dysfunction, subclinical vascular damage, systemic inflammation, and end-organ damage. Abnormalities induced in the HDL structure in both T1DM and T2DM that directly impair functionality involve both the lipidome and proteome and are augmented by the dysregulation of glucose and lipid metabolism, nonenzymatic glycation, and oxidative processes. Structure–function studies of HDL in hyperglycemic, dyslipidemic T2DM patients revealed both a gain and loss of lipidomic and proteomic components. Such changes attenuated both the optimal protective effects of HDL on mitochondrial function and its capacity to inhibit endothelial cell apoptosis, as well as its role in endothelial protection and vasodilatory activity. There is extensive evidence that both the proteome and lipidome of HDL are altered in T1DM and T2DM, with the impairment of multiple functions [45,46]. Despite the contrasting quantitative changes in HDL in T1DM, as compared with T2DM, the current data attest to the impaired functionality of HDL particles in both disorders. Glycemic burden is a principal factor in ED in both T1DM and T2DM; this effect is amplified in T2DM because of atherogenic dyslipidemia [47]. HDL dysfunctionality is, therefore, gaining traction as an innovative therapeutic target for the prevention of the premature, accelerated vascular disease typical of both T1DM and T2DM. 

Whether impaired HDL function plays a causal role in the pathophysiology of premature atherosclerotic cardiovascular disease in T1DM and T2DM requires further investigation. The development of ‘smart’ reconstituted HDL particles transporting the apoM/sphingosine-1-phosphate complex represents a novel therapeutic option for protection against the development of insulin resistance and vascular damage in diabetes [48].

## 7. Impact of Supplements on ED in DM

Several nutritional components have been studied, and their impact on endothelial function was demonstrated to be either positive (citrulline and glutathione [49]) or negative (α-ketoglutarate [50]).

Olive oil and omega-3 fatty acids are widely used supplements worldwide. Monounsaturated and polyunsaturated fatty acids improve postprandial metabolic control in healthy subjects and patients with T2DM [51]. In vitro studies have demonstrated that oleic acid reduces the expression of inflammatory adhesion molecules by endothelial cells, and polyphenols stimulate the synthesis and release of NO and protect the endothelium from post-prandial glucose-induced oxidative stress [51,52]. Conversely, the detrimental effect on endothelial function after a butter-enriched test meal could be caused by the downregulation of NO mediated by saturated fatty acids [53,54]. Cutruzolla et al. published a study that evaluated whether extra virgin olive oil (EVOO) and butter influence endothelial function in subjects with T1DM when added to a single high glycemic index (HGI) meal [55]. The primary goal of the study was to evaluate FMD after two different HGI meals enriched with EVOO and butter. Patients with T1DM had significantly lower endothelium-mediated vasodilation compared with non-diabetic controls. The authors observed a significant FMD increase over time after EVOO compared with butter in T1D. In control subjects, the endothelial function was not influenced by EVOO or butter. Blood glucose was significantly lower after EVOO than butter in T1DM, but it was almost unchanged in the control subjects. EVOO enhanced the vasodilatory capacity of the brachial artery via FMD examination compared with butter in patients with T1DM. The mechanisms by which EVOO and butter exert their effects are probably mediated by the intrinsic properties of the fats, thus influencing the paracrine activity of the ECs.

Omega-3 fatty acids are polyunsaturated fatty acids, such as eicosapentaenoic acid (EPA) and docosahexaenoic acid (DHA), and are contained in oily fish and the livers of white fish. They possess multiple biological properties including antioxidant, anti-inflammatory, immunomodulatory, antitumor, antidepressant, antihypertensive, and lipid-lowering effects [56]. Moreover, a growing body of evidence has revealed that various kinds of metabolic disorders underlying the development of diabetes may be ameliorated by omega-3 supplementation [57]. Omega-3 fatty acids can also be effective in improving vascular endothelial function by reducing inflammatory cytokines and increasing NO and oxylipin production [58,59].

Khorshidi et al. published a randomized, double-blind, placebo-controlled trial with 51 adolescents with T1DM who received either 600 mg per day of omega-3 fatty acids (containing 180 mg of EPA and 120 mg of DHA) or placebos for 12 weeks [60]. At baseline and after the intervention, the subjects were assessed via ultrasound and using biochemical parameters, including FMD, cIMT, high-sensitivity C-reactive protein (hs-CRP), erythrocyte sedimentation rates, triglycerides, LDL, high-density lipoproteins (HDLs), total cholesterol, blood urea nitrogen, creatinine, fasting blood glucose, HbA1c, and the HOMA-IR index.

It has been reported that FMD predicts vascular disease in adulthood and is impaired in adolescents with T1DM. This study revealed that daily supplementation with 600 mg of omega-3 for 12 weeks increased FMD, which was significantly correlated with lower triglyceride levels. Other parameters were not affected. The authors concluded that regular and sufficient omega-3 intake may improve vascular function, which may also attenuate the risk of CVD in adolescents with T1DM.

## 8. Is It Possible to Prevent ED in T1DM?

An interesting animal study evaluated the combined effect of endoplasmatic reticulum (ER) stress inhibition with angiotensin-converting enzyme 2 (ACE2) activation—which are two major contributors to hyperglycemia-induced ED—in preventing ED associated with T1DM [61]. Persistent hyperglycemia damages the endothelial layer via multiple signaling pathways, including enhanced oxidative stress, the downregulation of angiotensin-converting enzyme 2 signaling, and the exacerbation of ER stress. ER stress could suppress endothelial NO synthase expression [62] and may induce EC apoptosis and increase oxidative stress, both responsible for ED.

The hypothesis of the Sankrityayan study [61] was that the simultaneous inhibition of ER stress and the activation of ACE2 using a novel combination of diminazene aceturate (ACE2 activator) and tauroursodeoxyxcholic acid (TUDCA, ER stress inhibitor) might produce better vasoprotection and, therefore, could prevent ED. Treatment with TUDCA effectively reduced plasma glucose levels, which was in agreement with previously published studies [63]; however, diminazene aceturate monotherapy did not reduce blood glucose. The combination therapy was found to significantly reduce blood glucose levels when compared with a diabetic control group. ACE2 activation using diminazene aceturate markedly reduced oxidative stress levels, as shown by improved glutathione and reduced malondialdehyde levels. Combination therapy was better than monotherapies in alleviating oxidative stress and elevating NO bioavailability. The study found that treatments with TUDCA improved the levels of NO in the aortas of diabetic rats. 

Diabetes usually leads to the deposition of excess collagen in the aorta. The study observed a significant reduction in collagen deposition in animals treated with the ACE2 activator and ER stress inhibitor. Hyperglycemia-induced elevation in angiotensin-II and consequential ER stress might be a factor in collagen deposition in the aorta.

The authors demonstrated that the simultaneous inhibition of ER stress and ACE2 restored endothelial function more significantly by reducing oxidative stress, improving NO bioavailability, and reducing collagen deposition.

## 9. Conclusions

ED plays a crucial role not only in arterial stiffness and microvascular reactivity but is also one of the major factors that contribute to cardiovascular morbidity and mortality in patients with T1DM and T2DM. Even though there is a lot of published data about the association between T2DM and ED, most studies, including our own pilot study, have focused on T2DM, so the evidence for T1DM is still lacking. More interventional studies focusing on the influence of ED in T1DM are urgently needed to potentially prevent the development of microvascular complications and, therefore, increase the quality of life of patients with this disease and reduce future morbidity and mortality in both types of DM.

## Figures and Tables

**Figure 1 ijms-24-10705-f001:**
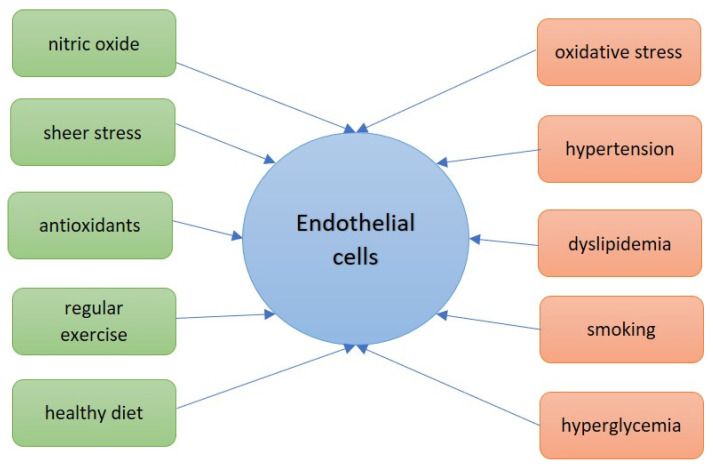
Factors that positively (green) and negatively (red) influence the endothelium.

**Figure 2 ijms-24-10705-f002:**
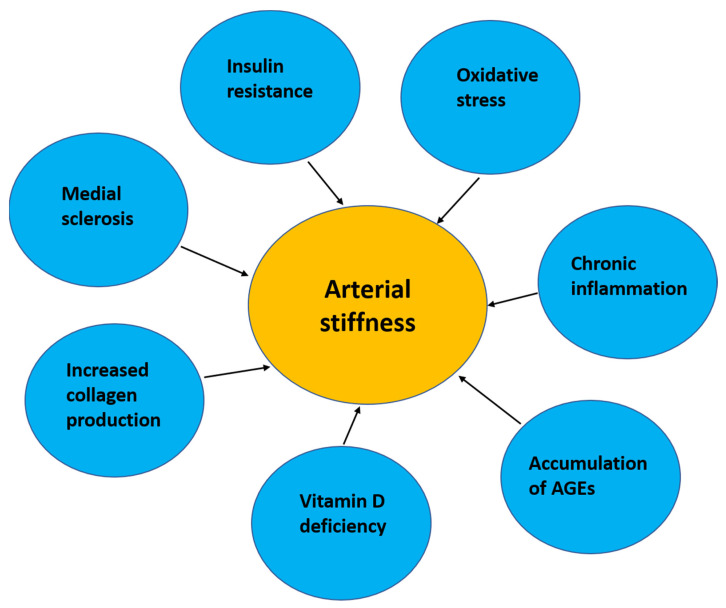
Factors influencing arterial stiffness.

## Data Availability

No data are available for this review article.

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
