# Peer review of "Endothelial Dysfunction in Diabetes Mellitus: New Insights"

_ijms, 2023, doi:10.3390/ijms241310705_

Round 1
Reviewer 1 Report
The authors, Dubsky M, et.al discussed the development of ED in diabetes and even the prevention of ED in T1DM and T2DM. It is a very interesting review article, with an update on this aspect. However, there are still some issues which should be addressed.
Authors may make the transitions smoother between sections and re-arrange the order of sections to facilitate reading.
Line96-100: the ED should be induced by the combination of hyperglycemia, insulin resistance and dyslipidemia, not only one or two factor(s).
Fig1, please add the illustration (fig Legend) of the immunohistochemistry; should ‘diabetes’ be changed to ‘hyperglycemia’?
MicroRNA part: Authors indicated miRs play important role. However only miR200-c was discussed.
Indicate the sources of figures (e.g., fig3, 4 and 5): are they published or not? If they are published, please show the refs.
EV part: how to define the EV/MV size? Authors discussed both miRs and EV. Can EV carry miRs?
EV was involved in ED in T1DM, as authors discussed. Does EV contribute to the development of ED in T2MD? What is/are the source(s) of EV? Is there any difference in the contribution to the development of ED between EVs from different sources?
In the part of Methods to assess ED: authors first talked about ‘arterial stiffness’. What is the relationship between ‘arterial stiffness’ and ED? Please describe it briefly.
In the section of ‘Our own experience with ED in T2DM’, authors discussed their own study. Please clearly describe if it is published or not.
Author Response
Dear Reviewer 1, thank you for your valuable comments, we provided a point-to-point response to your comments and suggestions, please see the attachment, they are marked in red font.
Best regards,
Michal Dubsky, corresponding author

Reviewer 2 Report
Dear authors,
The presented manuscript raises important questions on the role of DM in the process of endothelial function disruption.
In general, the manuscript is well prepared, but the revision of the language style used is suggested. At the moment, it is more pop-scientific rather than scientific.
Second of all, the section on the own experience should be removed. It is material for another original article.
Minor corrections:
1) What is the picture in the middle? I suggest removing it and making a classic "spider" chart instead.
2) Line 148: There are many methods aimed at evaluating arterial stiffness (AS).
- FMD is dedicated to measuring endothelial vasodilatory properties, IMT is dedicated to the assessment of vascular wall rebuild. Those are endothelial parameters, but not AS parameters.
3) Endo-PAT is the name of the tool, not the measurement. Please verify as you use RHI and Endo-PAT as synonyms.
4) Line 181-182: Moreover, they have shown that T2DM correlates strongly and positively with progressive stiffening of central rather than peripheral arteries.
- This reference does not justify such a statement, what is more, they examine the impact of the drug, not the DM itself. Another thing is that this statement is wrong - in PAD patients with DM, the arterial walls are highly calcified, mostly the peripheral ones (BTK and foot arteries).
5) The role of angiogenic factors is not sufficiently described. There are some new articles on the circulating angiogenic factors that should be read by the authors and the section regarding angiogenic endothelial functions should be expanded (i.ex.:https://doi.org/10.3390/biomedicines11061559)
Kind regards
Grammar and spelling are correct, but the style used is in my opinion pop-scientifc rather than scientific.
Author Response
Dear Reviewer 2, thank you for your valuable comments, we provided a point-to-point response to your comments and suggestions, please see the attachment, they are marked in red font.
Best regards,
Michal Dubsky, corresponding author

Round 2
Reviewer 1 Report
No further comments
Reviewer 2 Report
Dear Authors,
Thank you for the corrections and improvements introduced. At present of the manuscript, I do not have any other questions or requests.
Congratulations on your work!
Kind regards